# FAIRNESS MITIGATION VIA A GEOMETRIC FRAMEWORK FOR FAIRNESS (GEOFFAIR)

## ABSTRACT

Fairness is a critical concern in Machine Learning, impacting its applications across domains. Existing fairness analyses often rely on complex mathematics, lacking of intuitive understanding. In this study, we introduce *GEOFFair*, a Geometric Framework for Fairness. It represents Machine Learning elements as vectors and sets, offering a more intuitive understanding of fairness related concepts. GEOFFair visualizes fairness mitigation techniques as vector projections, it provides a solid base to investigate the bias injection, aiding in constructing proofs, and it enables the study of fairness properties by means of geometric considerations. The main contribution of the work is to highlight GEOFFair's effectiveness in fairness studies, demonstrating that solely maximizing accuracy based on observed labels may not always be optimal for fairness.

## 1 INTRODUCTION

Fairness concerns within the realm of machine learning (ML) have recently emerged as a prominent and critical challenge, pitching a shadow over the widespread adoption of data-driven AI in critical domains such as healthcare, economics, welfare, and policy-making Mehrabi et al. (2021).

Usually, these concerns are approached from a mathematical standpoint presenting challenging complexities, as it necessitates scuffling with statistical distributions and, in some cases, non-linear models Srivastava et al. (2019). Many of the existing analyses rely on sophisticated methodologies, which may prove less than intuitive for an exhaustive comprehension of the components involved in addressing fairness.

We argue that the field could benefit from a streamlined framework offering an intuitive and sound grasp of fundamental fairness concepts and mechanisms in the realm of AI. Accordingly, in this work we present GEOFFair, a *Geometric Framework for Fairness*, that casts distributions, functions (such as ML models), fairness constraints, and hypothesis spaces into vectors and sets. The main advantage of geometric frameworks lies in their capacity for visualization, facilitating insights into both data and model behaviour.

Our motivation for adopting this approach stems from successful applications in other facets of ML Kansizoglou et al. (2021); Bronstein et al. (2017), wherein the mapping of models into vector spaces, often achieved by concatenating their parameters, has simplified their representation and analysis. Through this lens, concepts like distance metrics, projections, similarities, and algorithms can be visualized to gain valuable insights Shahmirzadi et al. (2019).

In this paper we showcase the practical utility of our framework by revisiting fairness mitigation techniques within the context of the geometric framework. By visualizing mitigation as vector movements within the space, we highlight properties and unveil the effects of actions underpinned by the mitigation process. We delve into bias injection and debiasing, two facets of the same coin, to underscore the analytical prowess afforded by our framework.

## 2 GEOFFAIR: A GEOMETRIC FRAMEWORK FOR FAIRNESS

The following section aims to introduce the GEOFFair formal framework. To achieve this goal, we will concentrate on two primary aspects. First, we introduce the key components of a typical setting

for studying fairness of Machine Learning models; then, in Subsection 2.2 we introduce a vector representation for some key stastistical concepts, which serves as the basis for the framework proper in Subsection 2.3. The vector representation enables us to formalize fairness concepts and metrics in a clear and precise mathematical language. Finally, we will discuss how these vector representations exist within the same space, providing a common basis for comparing and contrasting different fairness metrics (Subsection 2.4).

## 2.1 Formalization of a Generic Fairness Problem

Traditionally, the task of learning fair representations has been always formulated using probability theory and statistical analysis. Within this context, both the input data and the ground truth are commonly represented as aleatory variables with their own probability distribution; machine learning models are then seen as parameterized functions over those distributions, whose aim is to minimize a measure of likelihood – i.e., the training loss; and, finally, fairness metrics are seen as functions operating on the conditional expectations of the variables. Before delving into the specifics and the key differences of our framework, let us recall the main concepts starting from the definition of our data.

Let $X = (X_1, \ldots, X_n)$ be a multivariate random variable with support $\mathcal{X}$ and distribution $P(X)$. This variable represents our input distribution and, in the context of learning fair representations, it must have one (or more) feature $X_i$ which is considered *protected*, namely it represents a sensitive attribute of the input against which we want to ensure a non-discriminative behaviour. Similarly, let $Y$ with support $\mathcal{Y}$ and distribution $P(Y)$ be another random variable, typically but not necessarily univariate. This represents our target distribution, i.e., the value on which we aim to forecast given the input distribution $X$.

In our analysis we focus on a supervised learning setting. This means that, at training time, we have information, typically in the form of a sample, on the joint probability $P(X, Y)$, which we can use to learn the machine learning model $\mathcal{M}$ that better approximates the conditional probability $P(Y \mid X)$. From this perspective, $\mathcal{M}$ is a function $f : \mathcal{X} \to \mathcal{Y}$.

While this is enough to formally represent the task of unconstrained machine learning, when taking into account fairness requirements we also need to introduce one or more predicates $\Pi$ defined over the conditional expectations of $X$ and $Y$; more specifically, we aim at maximizing the likelihood subject to the constraint $\Pi(X, Y)$. This predicates are often based on a divergence metric $K(\cdot)$ which measures the difference between the conditional distributions of the target variable $Y$ with respect to all those values that the protected feature $X_i$ could assume. For example, if $X_i \sim \{0, 1\}$, then a valid predicate could be:

$$\Pi(X, Y) = K(Y \mid X_i = 0, Y \mid X_i = 1) < t$$

In the context of learning fair representations, $K$ is referred to as a *fairness metric* and $t$ is a threshold applied to the reported unfairness level. To avoid excess of notation, in the rest of the paper we will use $K(Y)$ to indicate that the fairness metric is applied to a random variable $Y$ according to the protected attributes of the input distribution $X$ to which it is paired to.

## 2.2 From Distribution to Vectors in the Space

Due to the complexity of such a statistical viewpoint on the fairness-aware learning task, the main purpose of our framework is to switch to a alternative representation designed to enhance clarity and to enable visualization of the underlying processes. As a first step is that direction, we introduce a vector-based representation for some key probabilistic concepts that can be used in our context without any significant loss of generality.

**Probability Distributions and Functions**  The main idea we rely on is to represent probability distributions to arbitrary precision via an *infinite sample*. Formally:

**Notation 1** (Probability Distributions)**.** *Given the two data distributions $X$ and $Y$, we encode them as a vector $(x, y) = \{x_i, y_i\}_{i=1}^n$, with $x_i, y_i \sim P(X, Y)$ and $n \to \infty$.*

Intuitively, $X$ represents an observable that may serve as the input for an ML model, while $Y$ represents the quantity (or class) to be estimated. The same representation can be applied for the

individual distributions of $X$ and $Y$, which are therefore denoted as $x$ and $y$. Our approach makes it particularly easy to represent functions over random variables (e.g. Machine Learning models evaluated over their input). Formally:

**Notation 2** (Functions)**.** *A deterministic function $f$ over $X$ and $Y$ can then be naturally viewed as a vector $f(x, y) = \{f(x_i, y_i)\}_{i=1}^n$ with $n \to \infty$, i.e. just the vector with the function evaluation over all the samples.*

Functions that depend only on $X$ or only on $Y$ are sub-cases of the above definitions and are respectively denoted as $f(x)$ and $f(y)$.

There are a few observations worth making. First, while we use the term "vector" for simplicity, our definitions are closer to functions that map an index $i$ to an object such as $x_i$ or $y_i$. In other words, $x$, $y$, $f(x)$, etc. can be thought of as points in a Hilbert space. Second, our representations are not exact, but they will be sufficient to approximate key statistical properties with arbitrarily high probability. Exact representations for distributions exist and are well known, e.g. the Probability Mass Function or Probability Density Function; however, they do not enable constructing a simple 1-1 mapping between components in the vector (e.g. $x_i$) and function evaluations (e.g. $f(x_i)$), which is instead trivial with our approach.

**Equivalence of Expectation Predicates**   Many of the existing fairness metrics are expressed in terms of (conditional) expectations, i.e. averages, or can be reduced in such a form. For example, assuming $X$ is a binary protected attribute, the DIDI metric from Aghaei et al. (2019) is defined in terms of the discrepancy between the global average outcome and the average outcome for each protected group, i.e. $|\mathbb{E}[Y \mid X = 0] - \mathbb{E}[Y]| + |\mathbb{E}[Y \mid X = 1] - \mathbb{E}[Y]|$. Statistical parity in classification, which advocates for similar probabilities of a positive outcome across all groups, can be defined as $|\mathbb{E}[Y \mid X = 0] - \mathbb{E}[Y \mid X = 1]|$, and so on. Intuitively, this means that many fairness constraints can be viewed as predicates over (conditional) expectations.

The sample expectation function, represented by $\mu(\cdot)$, tends to converge towards the true expectation $\mathbb{E}[\cdot]$ as the sample size grows: we use this result to establish a form of equivalence between predicates expressed over a distribution and those expressed over a sample.

**Theorem 1.** *Let $\Pi(X, Y)$ be a predicate over (conditional) expectations for $X$ and $Y$ and let $\pi(\{x_i\}_{i=1}^n, \{y_i\}_{i=1}^n)$ be its sample counterpart. Then we have that:*

$$P\left(\Pi(X, Y) \Leftrightarrow \lim_{n \to \infty} \pi(\{x_i\}_{i=1}^n, \{y_i\}_{i=1}^n)\right) = 1 \tag{1}$$

*i.e. the two predicates are equivalent almost surely as the sample size grows if the involved expectations are finite.*

*Proof.* The two predicates are identical except for the use of the true and sample expectations. For the sake of simplicity and without loss of generality, let us assume the involved expectations are respectively $\mathbb{E}[Y]$ and $\mu(\{y_i\}_{i=1}^n)$. Since the samples are drawn independently from the same distribution, due to the strong law of large numbers we have that:

$$P\left(\lim_{n \to \infty} \mu(\{y_i\}_{i=1}^n) = \mathbb{E}[Y]\right) = 1 \tag{2}$$

Equivalence of the sample and true expectations then implies equivalence of $\Pi$ and $\pi$.   □

Notation 1 and Notation 2 give us the ability to transition from the conventional distribution paradigm of ML to the realm of vector spaces. Theorem 1 enables reasoning over the vector representation and translates almost certainly any result to the original distribution, at least as far as fairness metrics are concerned. Together, these tools allow us to leverage the power and interpretability of vector space representations in the context of fairness metrics, expanding the scope of analysis and decision-making.

## 2.3   THE FORMAL MODEL

As mentioned in Section 2.2, we focus on a supervised learning setting where the goal is to learn a model that maps inputs (always observable) to outputs (observable at training time and to be

estimated at inference time). In this context, we introduce four key mathematical objects that play a major role in the analysis of fairness issues in AI.

We represent the *infinite input distribution* by means of an *infinite input vector* $x_n \in \mathcal{X}^n$, with $n \to \infty$, according to Notation 1. For consistency reason with the following notation, we identify the *infinite input vector* as $x_\infty \in \mathcal{X}^\infty$. Concerning the output, we make a distinction between the distribution that can actually be observed and the one that we ideally wish to estimate. We start by introducing the following concept:

**Definition 1** (Ground Vector). *A ground vector $y_\infty^+ \in \mathcal{Y}^\infty$ represents data that can be observed and used as ground truth to learn machine learning models. It is paired with the input vector $x$.*

As inspired by Dutta et al. (2020), we model the fact that the ground truth might be subject to systemic social biases, but with a key difference. That is, we directly define an "unbiased" output vector rather than an unbiased input matrix, as our framework allows us to reason in terms of vector components within the output space.

**Definition 2** (Gold Vector & Biased Mapping). *A gold vector $y_\infty^* \in \mathcal{Y}^\infty$ represents the "unbiased" data obtained by sampling the* fair *output distribution before it is corrupted by social biases; accordingly, we can derive a ground vector $y_\infty^+$ by considering the application of a biased mapping over the gold vector, i.e.:*

$$y_\infty^+ = b_\infty(y_\infty^*), \tag{3}$$

*where $b_\infty^{x_\infty} : \mathcal{Y}^\infty \to \mathcal{Y}^\infty$ is called "biased mapping" and takes the input vector as parameter. Since we only work in the output space, $x$ is required to compute the fairness measure but can be considered as a constant for the purpose of our framework. For this reason, we use the compact notation $b_\infty$.*

Note that in practical applications, the gold vector is typically unobservable and therefore not accessible at training time. Still, explicitly modeling the unbiased distributions allows us to study in deeper detail the interplay between bias and fairness constraints.

In our framework, an ML model can be viewed as a function that maps input to output data. In supervised learning, the training process is typically viewed as that of selecting one model out of a pool of candidates, so as to minimize a loss metric. Formally, training amounts to solving in an exact or approximate fashion:

$$\arg\min_{f \in \mathcal{F}} \mathcal{L}(f(x), y_\infty^+) \tag{4}$$

where $f$ is the ML model, $\mathcal{L}$ is the chosen loss metric and $\mathcal{F}$ represents the set of possible models, usually defined by specifying an architecture (e.g. a number and size of layers in a feed-forward neural network, number of estimators and maximum depth in a random forest).

In our framework, however, the input vector $x_\infty$ is by construction fixed, thus making the model output the only relevant factor. In other words, two models are equivalent as long as they have the same output. This observation allows us to introduce a simplified representation of the classical notion of hypothesis space.

**Definition 3** (Hypothesis Space). *The hypothesis space $\hat{\mathbb{Y}}_\infty$ is the set of possible infinite-dimensional outputs for the chosen class of ML models, i.e.*

$$\hat{\mathbb{Y}}_\infty = \{y \in \mathcal{Y}^\infty \mid \exists f \in \mathcal{F}_\infty : f(x) = y\}$$

Intuitively, the hypothesis space can be viewed as the set of possible model outputs for the considered sample. A linear regression model will have a limited hypothesis space due to its ability to represent linear relationships only, while more complex models such as random forests and neural networks will have a much larger hypothesis space.

Finally, as we are considering a fairness scenario, we need to model a final mathematical object in order to guarantee a proper analysis of the phenomenon, namely the region in the output space that is considered fair.

**Definition 4** (Fair Space). *Let $\overline{\mathbb{Y}}_\infty \subseteq \mathcal{Y}^\infty$ be the set containing all the infinite-dimensional output vectors that are aligned with the fairness requirements.*

We make no assumption on the mathematical definition of the fair space. Nonetheless, it is worth noting that in many practical cases, this set is defined by means of a threshold $t$ on a fairness metric $K$, i.e. $\overline{\mathbb{Y}}_\infty = \{y \in \mathcal{Y}^\infty | K(y) \leq t\}$.

Once all the elements are defined, we can examine how they interact with each other. In the most general setup, we can not make any assumption about the relationships between $y_\infty^*$, $y_\infty^+$, $\hat{\mathbb{Y}}_\infty$, and $\overline{\mathbb{Y}}_\infty$. Without specific contextual information on data, models, and constraints, the relationships between these entities can vary significantly.

It is worth mentioning that, in the defined framework, all vectors and sets we introduced exist in the same space, which facilitates easy visualization (see Figures in Section 3). This visual representation can assist with proof-by-witness, allowing us to analyze and demonstrate relationships between these vectors more effectively.

### 2.4 RELATIONSHIPS BETWEEN ELEMENTS

Table 1: Possible one-to-one relationships. When comparing the two sets, we use $\pitchfork$ and $\cap$ as aliases for $\hat{\mathbb{Y}}_\infty \cap \overline{\mathbb{Y}}_\infty = \emptyset$ and $\hat{\mathbb{Y}}_\infty \cap \overline{\mathbb{Y}}_\infty \neq \emptyset$, $\hat{\mathbb{Y}}_\infty, \overline{\mathbb{Y}}_\infty$.

| | $\hat{\mathbb{Y}}_\infty$ | $\overline{\mathbb{Y}}_\infty$ | $y_\infty^+$ | $y_\infty^*$ |
|---|---|---|---|---|
| $\hat{\mathbb{Y}}_\infty$ | | $\pitchfork, \subseteq, \supseteq, \cap$ | $\ni, \not\ni$ | $\ni, \not\ni$ |
| $\overline{\mathbb{Y}}_\infty$ | | | $\ni, \not\ni$ | $\ni, \not\ni$ |
| $y_\infty^+$ | | | | $\equiv, \not\equiv$ |
| $y_\infty^*$ | | | | |

It is worth noting that the relationships among the objects in the framework allow us to establish interesting properties and interactions among the objects involved in fairness mitigation techniques. A full discussion is beyond the scope of this contribution; however, a summary of every potential one-to-one relationship is presented in Table 1. The complete list is provided in Appendix A. While the number of possible combinations is not small, it is nevertheless finite, which can be helpful for proving universally quantified statements (i.e., $\forall$ and $\nexists$). While we do not examine each possible scenario, it is worth to highlight some very common or interesting cases.

- If $\hat{\mathbb{Y}}_\infty \subseteq \overline{\mathbb{Y}}_\infty$, the machine learning model is said to be *fair-by-design* Nurock et al. (2021). While achieving this is challenging in many practical cases, it can be attained by incorporating explicit rules into the model, ensuring that certain deontological fair principles are always upheld.

- If $\hat{\mathbb{Y}}_\infty \supseteq \overline{\mathbb{Y}}_\infty$, the machine learning models can cover all existing fair outputs. This can be the case when employing powerful models like large neural networks.

- If $y_\infty^+ \in \hat{\mathbb{Y}}_\infty$, it can be perfectly represented by the machine learning model, although this representation is not guaranteed to be fair unless $y_\infty^+$ is already in the Fair Space. Conversely, when the model lacks the capacity to represent $y_\infty^+$ adequately, it will be trained to minimize the loss $\mathcal{L}$ between the labels and the model outputs. The same considerations applies also to the relationship between $y_\infty^*$ and $\hat{\mathbb{Y}}_\infty$. The only difference is that, in this case, the analysis is purely theoretical since no model can be trained on $y_\infty^*$, which is not observable in real-world scenarios.

- If $y_\infty^* \notin \overline{\mathbb{Y}}_\infty$, the fairness metric is not aligned with the true distribution and/or the threshold is too small, then searching for a fairer vector is an ill posed problem.

### 2.5 FINITE SAMPLES

All the possible applications and advantages (i.e. visualization) introduced by adopting the geometric framework require that the previous considerations hold also in a real-world scenario. To understand how we can apply *GEOFFair* to a finite dataset, we need to add a specific notation to

differentiate a real case from the ideal one. Thus, we represent the *finite input distribution* by means of a *finite input vector* $x_n \in \mathcal{X}^n$, with $n \in \mathbb{N}$.

Concerning the output, we make a distinction between the distribution that can actually be observed and the one that we ideally wish to estimate. Moreover, we highlight that the notations refers to finite vectors. Thus, we start by introducing a new concept:

**Definition 5** (Finite Vectors). *Given a ground vector $y_\infty^+ \in \mathcal{Y}^\infty$ and the respective gold vector $y_\infty^* \in \mathcal{Y}^\infty$ both for $n \to \infty$, we analogously define the finite ground vector $y_n^+ \in \mathcal{Y}^n$ and the respective finite gold vector $y_n^* \in \mathcal{Y}^n$ for $n \in \mathbb{N}$. It still holds the same relationship of the infinite case:*

$$y_n^+ = b(y_n^*), \quad \text{where } b_n : \mathcal{Y}^n \to \mathcal{Y}^n \text{ is called finite biased mapping}$$

We can demonstrate that when $n$ increases, the statistical and analytical properties of the data tend to converge to the ones we obtained in the infinite case. In other words, for a finite value of $n$, we are performing a *sampling* of a continuous distribution which implies the validity of the transition from the conventional distribution paradigm of ML to the realm of vector spaces is no more guaranteed. In the following section we will show for as many statements as possible that the considerations assuming $n \to \infty$ can be generalized also to the finite samples scenario.

**Definition 6** (Finite Spaces). *As done for the ground vector and the gold vector, we can define the finite Hypothesis Space $\hat{\mathbb{Y}}_n \subseteq \mathcal{Y}^n$ for $n \in \mathbb{N}$ as the set of possible finite-dimensional outputs for the chosen class of ML models, i.e.*

$$\hat{\mathbb{Y}}_n = \{y \in \mathcal{Y}^n \mid \exists f \in \mathcal{F}_n : f(x) = y\}$$

*Finally, let $\overline{\mathbb{Y}}_n \subseteq \mathcal{Y}^n$ for $n \in \mathbb{N}$ be the set containing all the finite-dimensional output vectors that are aligned with the fairness requirements called* finite Fair Space.

## 3 FAIRNESS MITIGATION THROUGH THE LENS OF GEOFFAIR

In this section, we will utilize the GEOFFair framework to analyze fairness mitigation techniques. In a previous work by Dutta et al. Dutta et al. (2020), it was demonstrated that maximizing accuracy solely based on the observed labels vector may not always be the optimal choice. They employed statistical distributions and mathematical tools from probability theory to establish this result. Rather than extending their findings, our objective is to employ our proposed geometric framework to support and validate them. By leveraging the GEOFFair framework, we aim to present similar conclusions in a more accessible and interpretable way and can bridge the gap between complex mathematical concepts and practical implications. This allows for a clearer comprehension of the challenges associated with fairness and the potential solutions that can be pursued.

First, we propose the use of projections applied to the finite-dimensional case where standard numerical techniques can be adopted to algorithmically find the solution, then we extend the properties in the ideal case of an infinite-dimensional space. Finally, we introduce the problem of data polarization both as *bias injection* in a synthetic case and *bias removal* in a real-world case.

### 3.1 MITIGATION AS PROJECTION FOR FINITE SAMPLES

Mitigation, in the AI fairness context, refers to the process of reducing unfairness by either transforming the biased distribution or by ensuring that the ML model behaviour is compatible with the fairness constraints. From a geometric point of view, such techniques can be viewed as projecting either the ground vector or the ML output onto the Fair Space. Analogously, training an ML model can be viewed as the problem of finding a vector in the Hypothesis Space that is closest to the ground vector in terms of the loss function, i.e. as projecting the ground vector onto the Hypothesis Space. Therefore, in the context of GEOFFair, projections provide a convenient lens through which we can study mitigation at pre-processing, training, and post-processing time in a uniform fashion.

We focus our analysis on the more widespread case where learning a fair ML model is possible (i.e. $\hat{\mathbb{Y}}_n \cap \overline{\mathbb{Y}}_n \neq \emptyset$). We start by introducing two additional vectors, i.e. the projections of the ground truths and the gold standard vector, respectively. These projections will be onto the intersection space between the Hypothesis and the Fair Space.

**Definition 7** (Ground and Gold Fair Projections). *The optimal fair predictions $p_n^+$ and $p_n^*$ obtained from the ground ($y_n^+$) and gold ($y_n^*$) vectors for $n \in \mathbb{N}$, i.e.:*

$$p_n^+ = \arg\min_v \{\mathcal{L}(v, y_n^+) \mid v \in \hat{\mathbb{Y}}_n \cap \overline{\mathbb{Y}}_n\} \tag{5}$$

$$p_n^* = \arg\min_v \{\mathcal{L}(v, y_n^*) \mid v \in \hat{\mathbb{Y}}_n \cap \overline{\mathbb{Y}}_n\} \tag{6}$$

Intuitively, $p_n^+$ represents the outcome of training an ML model under fairness constraints, or equivalently of training an ML model over a ground distribution transformed so as to enforce the fairness restrictions. The $p_n^*$ vector represents the best fair model that we could learn for the (typically unobservable) "unbiased" distribution.

It is worth noting that $p_n^+$ and $p_n^*$ might not be unique, as equally accurate outputs that are both fair and representable by the model can exist. Moreover, the finite gold vector $y_n^*$ is inherently not unique since all the vectors obtained by sampling the *fair distribution* are, by definition, gold vectors; this trivially leads to multiple projections $p_n^*$. Furthermore, for the purpose of our theoretical analysis, we will assume that $p_n^+$ and $p_n^*$ are obtained from exact and globally optimal algorithms. However, it is important to acknowledge that many machine learning models, especially larger ones, do not guarantee this optimality property in practice. Additionally, to avoid trivial cases, we assume that the *biased mapping* function $b_n: \mathcal{Y}^n \to \mathcal{Y}^n$ applies a modification to the input vector, i.e. that $y_n^* \not\equiv y_n^+$. This assumption narrows down our analysis to even fewer cases than those defined in Subsection 2.4, and let us draw the following conclusion:

$$\mathcal{L}(y_n^+, y_n^*) > 0 \tag{7}$$

where $\mathcal{L}$ is any non-negative loss function such that $\mathcal{L}(y_n^+, y_n^*) = 0$ iff $y_n^+ \equiv y_n^*$.

**Basic Properties of Fair Projections**  Let us consider the optimization problems defined in Equations (5)-(6) and examine the behaviour of $p_n^+$ and $p_n^*$ in terms of fairness based on the position of $y_n^+$ and $y_n^*$, respectively. We will rely on the formulation of the Fair Space based on a fairness metric $K(\cdot)$ that we introduced in Section 2, i.e.:

$$\overline{\mathbb{Y}}_n = \{y \in \mathcal{Y}^n \mid K(y) \leq t\} \tag{8}$$

**Property 1** (Fair Projections). *Given a vector $y$ and its projection $y'$ onto the Fair Space as defined in Equation (8), we know that:*

$$y \in \overline{\mathbb{Y}}_n \implies y' \equiv y \implies K(y') = K(y) \tag{9}$$

$$y \notin \overline{\mathbb{Y}}_n \implies K(y') = t \tag{10}$$

*meaning that any vector lying within the Fair Space will be projected onto itself (thus exhibiting the same fairness level); conversely, if the vector is outside the Fair Space, its projection will be on the boundary of the Fair Space, resulting in threshold-level fairness.*

This is a well-known property in both convex and non-convex optimization, whose proof can be found in Jain & Kar (2017). Now, if we take into account the capabilities of the ML model, we can extend Property 1 as follows:

**Property 2** (Representable Fair Projections). *Given a vector $y$ and its projection $y'$ onto the intersection between the Fair and Hypothesis Space, we know that:*

$$y \in \overline{\mathbb{Y}}_n \vee \hat{\mathbb{Y}}_n \subseteq \overline{\mathbb{Y}}_n \implies K(y') \leq t \tag{11}$$

$$y \notin \overline{\mathbb{Y}}_n \wedge \hat{\mathbb{Y}}_n \supseteq \overline{\mathbb{Y}}_n \implies K(y') = t \tag{12}$$

It is important to note that when the Fair Space and the Hypothesis Space have a non-trivial intersection – i.e. neither space is a subset of the other –, we cannot draw conclusions about $K(y')$ since points in the boundary of the intersection can exhibit different fairness levels.

### 3.2 MITIGATION AS PROJECTION FOR INFINITE SAMPLES

All the considerations made in the previous paragraph can be generalized for the infinite-dimensional case with $n \to \infty$. In particular, we define $p_\infty^+$ and $p_\infty^*$ in the same way as Equations 5 and 6

respectively. One of the main differences lies between $y_n^*$ and $y_\infty^*$: in the finite-dimensional case it is easy to conclude that there can be multiple gold vectors due to the distribution sampling operation, each generated vector is probably close to the others in the output space (for example considering the loss function as distance measure) but we can assert that is almost impossible for all the existing gold vectors to coincide in the very same vector.

**Property 3** (Multiple finite gold vectors). *Given a set of m finite-dimensional gold vectors sampled from the same fair distribution, the probability that at least two of them do not coincide is 1, i.e.:*

$$\lim_{m \to \infty} P\Big(\exists i, j \mid 0 \le i < j < m \land \mathcal{L}\big(\{y_n^*\}_i, \{y_n^*\}_j\big) > 0\Big) = 1 \tag{13}$$

In the realm of infinite-dimensional samples, a different phenomenon emerges: the distance between any two gold vectors tends to zero, as they nearly perfectly represent the same distribution, as demonstrated in Theorem 1.

**Property 4** (Multiple infinite gold vectors). *Given a set of m infinite-dimensional gold vectors sampled from the same fair distribution, the probability that at least two of them do not coincide is 0, i.e.:*

$$P\Big(\exists i, j \mid 0 \le i < j < m \land \mathcal{L}\big(\{y_\infty^*\}_i, \{y_\infty^*\}_j\big) > 0\Big) = 0 \quad \forall m \in [2, \infty) \tag{14}$$

*Using the notation $y_\infty^*$, this definition implies a $\lim n \to \infty$. Finally, as a consequence of the fact that the mutual distance between any pair $(\{y_\infty^*\}_i, \{y_\infty^*\}_j)$ goes to zero, we can conclude that it exists a vector $u_\infty^*$ towards all the gold vectors converge.*

Studying the properties of $u_\infty^*$ allows us to make considerations for a single vector and then generalise them to any gold vector introducing an arbitrary small error. Properties 1 and 2 are still valid in a Hilbert Space, keeping substantially unchanged the considerations made about the value of $K(\cdot)$ for the projections with respect to the threshold $t$.

We can assess that the mitigation process remains substantially unchanged in the infinite-dimensional cases despite the different properties of the gold vector.

### 3.3 Injecting Bias – Polarization

Throughout the whole paper, we have consistently emphasized the distinction between two crucial elements: the gold vector, a *n*-dimensional sample of the target fair distribution, and the ground vector, the real-world data derived from the gold vector by means of a *biased mapping b* (Definition 5). This biasing effect is often linked to distortions resulting from the data collection process, such as imbalances in the population, or even deliberate unfair practices driven by human bias or a combination of both factors. Consequently, it is quite natural to think about how to solve the challenge of reversing the biased mapping to retrieve the gold vector.b However, it is worth mentioning that we can also explore a complementary perspective on this issue. Instead of unbiasing the model, we can investigate methods to intentionally introduce controlled bias into a given gold vector, thereby generating a desired ground vector. This operation of "controlled bias injection" can be referred to as *polarization*.

The geometric framework we illustrated in the previous sections not only offers an intuitive visual comprehension of *why* a biased mapping significantly impacts an ML model's ability to attain an optimal and fair solution but also may provide insights *how* is possible to polarize a dataset (more specifically the target feature) to obtain a certain configuration between the ground vector and the Fair o the Hypothesis Spaces. This type of investigation can be conducted using synthetic datasets, which offer the capability to deliberately introduce controlled, arbitrary biases. This approach allows for the precise manipulation of both the distance and the orientation of the newly perturbed vector in relation to the original one.

In the highly simplified example shown in Figure 1, we can intuitively understand the relevance of controlling the process of bias injection and its pivotal role in examining the interplay between the ground vector and the gold vector in a geometric sense. It's worth noting that in this example, we solely assess which segment of the output space contains a particular entity. We do not take into account the effectiveness or the quality of the mitigation; instead, our focus is solely on evaluating the ML model's ability to predict the projected vector while adhering to the fairness constraint.

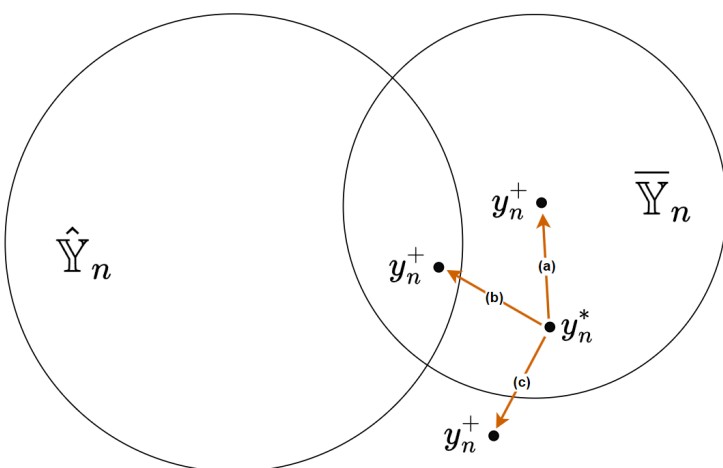

Figure 1: The picture illustrates three possible results of a polarization applied to the same gold vector (for simplicity reasons, we assume that $y_n^*$ is unique). In this example the gold vector belongs to the Fair Space but not to the Hypothesis Space. In **(a)** we notice that the biased mapping produces a different vector, but the same relationships with the other entities of the original one still hold. In **(b)** the biased mapping eases the problem since the ground vector belongs not only to the Fair Space but also to the Hypothesis Space (becoming a valid output for an ML model). Finally, case **(c)** leads to the worst scenario where the ground vector still can not be returned by an ML model, but now the fairness constraint is violated as well. It's important to emphasize that the three depicted polarizations share an equal magnitude (in terms of the Euclidean distance), yet merely altering the direction of the biased mapping leads to significant disparities.

**Possible applications**  Creating a synthetic dataset with well-defined properties is a common challenge that developers and researchers have to address in many ML applications. Among all the possible examples which demonstrate the large employment of generated data, we can mention the what-if analyses where emulating specific conditions is required to perform experiments, or we can even think about the benchmarking and evaluation phase of a traditional unbiasing technique; both these cases can rarely rely on extensive and representative historical datasets. Thanks to the polarization process, it is possible to tune the alteration of the biased target feature in order to generate a problem which can result particularly demanding for a certain mitigation technique, showing the weakness (or the strengths) of the approach that needs to be validated on a wide number of scenarios.

Therefore, even when historical data are available it might be convenient to tackle the problem using properly generated synthetic datasets. Highly replicability on different domains allows to better generalize the quality of a novel methodology, in fact the polarization technique can quite intuitively show when a injected bias is actually moving a ground vector further from its projection or it is just producing equivalent vectors according to the entities we defined, and if the belonging of a vector to its original output segment is changing. This interesting double point of view on polarization (studying how to retrieve the gold vector, and enforcing the validation procedure of already existent techniques) highlight not only the agnostic nature of this framework but also one of the most salient and direct applications to real-world use cases.

## 4  CONCLUSION

This study has introduced GEOFFair, a novel GEOmetric Framework for Fairness, which harnesses geometric principles to offer a robust and intuitive comprehension of fairness in the realm of AI. We have highlighted the benefits of adopting such a geometric framework for addressing fairness concerns. Furthermore, our analysis in this study has employed GEOFFair to examine fairness mitigation strategies and bias injection, ultimately yielding polarized datasets that serve as valuable tools for assessing and testing fairness in AI systems.

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

## A   APPENDIX: POSSIBLE SCENARIOS

Let $\mathbb{Z}$ be $\mathcal{Y} \setminus (\hat{\mathbb{Y}}_\infty \cup \overline{\mathbb{Y}}_\infty)$.

SCENARIO $\hat{\mathbb{Y}}_\infty \cap \overline{\mathbb{Y}}_\infty = \emptyset$

Cases in which $y* \in \overline{\mathbb{Y}}_\infty$:

    1) $y_\infty^*, y_\infty^+ \in \overline{\mathbb{Y}}_\infty$ and $y_\infty^* \equiv y_\infty^+$.

    2) $y_\infty^*, y_\infty^+ \in \overline{\mathbb{Y}}_\infty$ and $y_\infty^* \not\equiv y_\infty^+$.

    3) $y_\infty^* \in \overline{\mathbb{Y}}_\infty$ and $y_\infty^+ \in \mathbb{Z}$.

    4) $y_\infty^* \in \overline{\mathbb{Y}}_\infty$ and $y_\infty^+ \in \hat{\mathbb{Y}}_\infty$.

Cases in which $y* \notin \overline{\mathbb{Y}}_\infty$:

    5) $y_\infty^*, y_\infty^+ \in \mathbb{Z}$ and $y_\infty^* \equiv y_\infty^+$.

    6) $y_\infty^*, y_\infty^+ \in \mathbb{Z}$ and $y_\infty^* \not\equiv y_\infty^+$.

    7) $y_\infty^* \in \mathbb{Z}$ and $y_\infty^+ \in \overline{\mathbb{Y}}_\infty$.

    8) $y_\infty^* \in \mathbb{Z}$ and $y_\infty^+ \in \hat{\mathbb{Y}}_\infty$.

    9) $y_\infty^*, y_\infty^+ \in \hat{\mathbb{Y}}_\infty$ and $y_\infty^* \equiv y_\infty^+$.

    10) $y_\infty^*, y_\infty^+ \in \hat{\mathbb{Y}}_\infty$ and $y_\infty^* \not\equiv y_\infty^+$.

    11) $y_\infty^* \in \hat{\mathbb{Y}}_\infty$ and $y_\infty^+ \in \mathbb{Z}$.

    12) $y_\infty^* \in \hat{\mathbb{Y}}_\infty$ and $y_\infty^+ \in \overline{\mathbb{Y}}_\infty$.

SCENARIO $\overline{\mathbb{Y}}_\infty \subset \hat{\mathbb{Y}}_\infty$

Cases in which $y* \in \overline{\mathbb{Y}}_\infty$:

    13) $y_\infty^*, y_\infty^+ \in \overline{\mathbb{Y}}_\infty$ and $y_\infty^* \equiv y_\infty^+$.

    14) $y_\infty^*, y_\infty^+ \in \overline{\mathbb{Y}}_\infty$ and $y_\infty^* \not\equiv y_\infty^+$.

    15) $y_\infty^* \in \overline{\mathbb{Y}}_\infty$ and $y_\infty^+ \in \hat{\mathbb{Y}}_\infty \setminus \overline{\mathbb{Y}}_\infty$.

    16) $y_\infty^* \in \overline{\mathbb{Y}}_\infty$ and $y_\infty^+ \in \mathbb{Z}$.

Cases in which $y* \notin \overline{\mathbb{Y}}_\infty$:

    17) $y_\infty^*, y_\infty^+ \in \hat{\mathbb{Y}}_\infty \setminus \overline{\mathbb{Y}}_\infty$ and $y_\infty^* \equiv y_\infty^+$.

    18) $y_\infty^*, y_\infty^+ \in \hat{\mathbb{Y}}_\infty \setminus \overline{\mathbb{Y}}_\infty$ and $y_\infty^* \not\equiv y_\infty^+$.

    19) $y_\infty^* \in \hat{\mathbb{Y}}_\infty \setminus \overline{\mathbb{Y}}_\infty$ and $y_\infty^+ \in \overline{\mathbb{Y}}_\infty$.

    20) $y_\infty^* \in \hat{\mathbb{Y}}_\infty \setminus \overline{\mathbb{Y}}_\infty$ and $y_\infty^+ \in \mathbb{Z}$.

    21) $y_\infty^*, y_\infty^+ \in \mathbb{Z}$ and $y_\infty^* \equiv y_\infty^+$.

    22) $y_\infty^*, y_\infty^+ \in \mathbb{Z}$ and $y_\infty^* \not\equiv y_\infty^+$.

    23) $y_\infty^* \in \mathbb{Z}$ and $y_\infty^+ \in \hat{\mathbb{Y}}_\infty \setminus \overline{\mathbb{Y}}_\infty$.

    24) $y_\infty^* \in \mathbb{Z}$ and $y_\infty^+ \in \overline{\mathbb{Y}}_\infty$.

SCENARIO $\hat{\mathbb{Y}}_\infty \subset \overline{\mathbb{Y}}_\infty$

Cases in which $y* \in \overline{\mathbb{Y}}_\infty$:

25) $y_\infty^*, y_\infty^+ \in \hat{\mathbb{Y}}_\infty$ and $y_\infty^* \equiv y_\infty^+$.

26) $y_\infty^*, y_\infty^+ \in \hat{\mathbb{Y}}_\infty$ and $y_\infty^* \not\equiv y_\infty^+$.

27) $y_\infty^* \in \hat{\mathbb{Y}}_\infty$ and $y_\infty^+ \in \overline{\mathbb{Y}}_\infty \setminus \hat{\mathbb{Y}}_\infty$.

28) $y_\infty^* \in \hat{\mathbb{Y}}_\infty$ and $y_\infty^+ \in \mathbb{Z}$.

29) $y_\infty^*, y_\infty^+ \in \overline{\mathbb{Y}}_\infty \setminus \hat{\mathbb{Y}}_\infty$ and $y_\infty^* \equiv y_\infty^+$.

30) $y_\infty^*, y_\infty^+ \in \overline{\mathbb{Y}}_\infty \setminus \hat{\mathbb{Y}}_\infty$ and $y_\infty^* \not\equiv y_\infty^+$.

31) $y_\infty^* \in \overline{\mathbb{Y}}_\infty \setminus \hat{\mathbb{Y}}_\infty$ and $y_\infty^+ \in \hat{\mathbb{Y}}_\infty$.

32) $y_\infty^* \in \overline{\mathbb{Y}}_\infty \setminus \hat{\mathbb{Y}}_\infty$ and $y_\infty^+ \in \mathbb{Z}$.

Cases in which $y* \notin \overline{\mathbb{Y}}_\infty$:

33) $y_\infty^*, y_\infty^+ \in \mathbb{Z}$ and $y_\infty^* \equiv y_\infty^+$.

34) $y_\infty^*, y_\infty^+ \in \mathbb{Z}$ and $y_\infty^* \not\equiv y_\infty^+$.

35) $y_\infty^* \in \mathbb{Z}$ and $y_\infty^+ \in \overline{\mathbb{Y}}_\infty \setminus \hat{\mathbb{Y}}_\infty$.

36) $y_\infty^* \in \mathbb{Z}$ and $y_\infty^+ \in \hat{\mathbb{Y}}_\infty$.

SCENARIO $\hat{\mathbb{Y}}_\infty \cap \overline{\mathbb{Y}}_\infty \neq \emptyset$ AND NOT A PREVIOUS CASE

Cases in which $y* \in \overline{\mathbb{Y}}_\infty$:

37) $y_\infty^*, y_\infty^+ \in \overline{\mathbb{Y}}_\infty \setminus \hat{\mathbb{Y}}_\infty$ and $y_\infty^* \equiv y_\infty^+$.

38) $y_\infty^*, y_\infty^+ \in \overline{\mathbb{Y}}_\infty \setminus \hat{\mathbb{Y}}_\infty$ and $y_\infty^* \not\equiv y_\infty^+$.

39) $y_\infty^* \in \overline{\mathbb{Y}}_\infty \setminus \hat{\mathbb{Y}}_\infty$ and $y_\infty^+ \in \overline{\mathbb{Y}}_\infty \cap \hat{\mathbb{Y}}_\infty$.

40) $y_\infty^* \in \overline{\mathbb{Y}}_\infty \setminus \hat{\mathbb{Y}}_\infty$ and $y_\infty^+ \in \hat{\mathbb{Y}}_\infty \setminus \overline{\mathbb{Y}}_\infty$.

41) $y_\infty^* \in \overline{\mathbb{Y}}_\infty \setminus \hat{\mathbb{Y}}_\infty$ and $y_\infty^+ \in \mathbb{Z}$.

42) $y_\infty^*, y_\infty^+ \in \overline{\mathbb{Y}}_\infty \cap \hat{\mathbb{Y}}_\infty$ and $y_\infty^* \equiv y_\infty^+$.

43) $y_\infty^*, y_\infty^+ \in \overline{\mathbb{Y}}_\infty \cap \hat{\mathbb{Y}}_\infty$ and $y_\infty^* \not\equiv y_\infty^+$.

44) $y_\infty^* \in \overline{\mathbb{Y}}_\infty \cap \hat{\mathbb{Y}}_\infty$ and $y_\infty^+ \in \hat{\mathbb{Y}}_\infty \setminus \overline{\mathbb{Y}}_\infty$.

45) $y_\infty^* \in \overline{\mathbb{Y}}_\infty \cap \hat{\mathbb{Y}}_\infty$ and $y_\infty^+ \in \overline{\mathbb{Y}}_\infty \setminus \hat{\mathbb{Y}}_\infty$.

46) $y_\infty^* \in \overline{\mathbb{Y}}_\infty \cap \hat{\mathbb{Y}}_\infty$ and $y_\infty^+ \in \mathbb{Z}$.

Cases in which $y* \notin \overline{\mathbb{Y}}_\infty$:

47) $y_\infty^*, y_\infty^+ \in \hat{\mathbb{Y}}_\infty \setminus \overline{\mathbb{Y}}_\infty$ and $y_\infty^* \equiv y_\infty^+$.

48) $y_\infty^*, y_\infty^+ \in \hat{\mathbb{Y}}_\infty \setminus \overline{\mathbb{Y}}_\infty$ and $y_\infty^* \not\equiv y_\infty^+$.

49) $y_\infty^* \in \hat{\mathbb{Y}}_\infty \setminus \overline{\mathbb{Y}}_\infty$ and $y_\infty^+ \in \overline{\mathbb{Y}}_\infty \cap \hat{\mathbb{Y}}_\infty$.

50) $y_\infty^* \in \hat{\mathbb{Y}}_\infty \setminus \overline{\mathbb{Y}}_\infty$ and $y_\infty^+ \in \overline{\mathbb{Y}}_\infty \setminus \hat{\mathbb{Y}}_\infty$.

51) $y_\infty^* \in \hat{\mathbb{Y}}_\infty \setminus \overline{\mathbb{Y}}_\infty$ and $y_\infty^+ \in \mathbb{Z}$.

52) $y_\infty^*, y_\infty^+ \in \mathbb{Z}$ and $y_\infty^* \equiv y_\infty^+$.

53) $y_\infty^*, y_\infty^+ \in \mathbb{Z}$ and $y_\infty^* \not\equiv y_\infty^+$.

54) $y_\infty^* \in \mathbb{Z}$ and $y_\infty^+ \in \hat{\mathbb{Y}}_\infty \setminus \overline{\mathbb{Y}}_\infty$.

55) $y_\infty^* \in \mathbb{Z}$ and $y_\infty^+ \in \overline{\mathbb{Y}}_\infty \cap \hat{\mathbb{Y}}_\infty$.

56) $y_\infty^* \in \mathbb{Z}$ and $y_\infty^+ \in \overline{\mathbb{Y}}_\infty \setminus \hat{\mathbb{Y}}_\infty$.

