# OpenReview forum: "GEOFFair: a GEOmetric Framework for Fairness"
_ICLR.cc/2024/Conference — Submitted to ICLR 2024_

### Official Review · Reviewer_xPkB · 2023-10-30

**Soundness:** 1 poor
**Presentation:** 1 poor
**Contribution:** 1 poor
**Rating:** 3
**Confidence:** 4

**Summary:**

This paper introduces GEOFFair, a framework that convert the conventional fairness formulation using probability distributions into vector representation in the Hilbert space. The authors claim that their formulation offers a more intuitive understanding of fairness related concepts, and visualizes fairness mitigation techniques.

**Strengths:**

1.	The authors provide a taxonomy of fairness in Section 2.4 under their vector representation
2.	The authors discuss the case of finite samples

**Weaknesses:**

1.	The authors claim that the conventional formulation for fairness interventions using probability distributions is challenging; however, they fail to provide concrete evidence on why it is challenging (e.g., in terms of computational efficiency). Therefore, their proposed vector representation seems unnecessary, as I do not see any advantages using this vector representation in terms of easier optimization, or could potentially lead to a new fairness metric with better operational meanings.
2.	The authors claim that the proposed vector representation offers a more intuitive understanding and visualization, but fails to provide any evidence throughout the paper (just one figure in Section 3.3), nor do they provide any algorithm to implement/ compute the Fair Projection stated in Property 1. There are also no numerical results at all.
3.	The taxonomy introduced in Section 2.4 is trivial and provide no new insights for fairness problems.
4.	The connection/ discussion between existing fairness intervention methods/ notions and the proposed vector representation is lacking, indicating an insufficient literature survey. For example, how could existing methods such as multi-calibration, multi-accuracy, FERMI, FairProjection, Reduction, etc. benefit from this new framework?
5.	The authors only discuss very limited fairness constraints (equalized odds, a very common metric, is missing), and it is doubtful that the proposed framework is meaningful for many other fairness constraints.
6.	Despite that the authors discuss finite sample case, they fail to provide a sample complexity bound/ error probability bounds.
7.	It is well-known that the accuracy and fairness forms a trade-off. This paper fails to discuss this trade-off, which is the most important consideration in practice.

**Questions:**

Please refer to the Weaknesses.

---

### Official Review · Reviewer_p9x9 · 2023-11-01

**Soundness:** 2 fair
**Presentation:** 2 fair
**Contribution:** 2 fair
**Rating:** 5
**Confidence:** 3

**Summary:**

In this paper, the authors introduce GEOFFair, a geometric framework that offers an intuitive and visual understanding of fairness in machine learning (ML). It represents ML components as vectors and sets, making fairness concepts and mitigation techniques easier to grasp and analyze. Specifically, the main contributions of GEOFFair include visualizing fairness mitigation as vector projections, investigating bias injection, constructing proofs, and studying fairness properties through geometric considerations. The authors emphasize that maximizing accuracy based on observed labels may not always lead to optimal fairness outcomes.

**Strengths:**

1.	In this paper, the authors introduces GEOFFair, a Geometric Framework for Fairness, which stands out as an innovative approach in the realm of machine learning fairness. By representing machine learning elements as vectors and sets, it provides a more intuitive and visual understanding of fairness-related concepts, which is a significant departure from traditional, complex mathematical analyses.

2.	 GEOFFair not only provides theoretical insights but also offers practical utility through the visualization of fairness mitigation techniques.

**Weaknesses:**

1.	The paper does not provide a clear and specific definition of 'fair outputs.' Given the potential for varied interpretations of fairness, a concrete definition or examples would enhance the manuscript's clarity and applicability. I recommend that the authors incorporate case studies or examples to illustrate the potential differences in fair outputs and how these differences can be quantified within the vector space.

2.	This paper lacks an in-depth discussion on the trade-off between model fairness and performance. This is a crucial aspect of machine learning models, and the manuscript would benefit from a detailed explanation on how to measure and visualize this trade-off. I suggest that the authors add content to explicitly address this trade-off, providing methodologies or visual aids to assist readers in understanding its implications on the model's outputs.

3.	Some of the assumptions made in this paper are not clearly articulated. Providing additional examples and a clearer explanation of these assumptions would aid in the reader’s comprehension and application of the presented framework. I recommend that the authors revisit the section on assumptions, ensuring that they are explicitly stated and supported by relevant examples.

4.	This paper should address how the labeling of assumptions in the proposed methodology is ensured to be unbiased. Additionally, given the possibility of biased labeling, the manuscript would benefit from a discussion on how to validate the model under such circumstances. I suggest that the authors provide guidelines or methods to check for biases, mitigate them, and validate the model.

**Questions:**

In the paper, the term 'fair outputs' is used extensively, but it lacks a specific definition or concrete examples. I appreciate if the authors could provide a clear and specific definition of 'fair outputs' and enhance the paper with case studies or examples that illustrate potential variations in fair outputs? How can these differences be quantified within the vector space to provide clarity and applicability to proposed framework?

In the paper, the authors touches on the crucial aspect of balancing model fairness and performance, but it does not delve into a comprehensive discussion or provide methodologies to measure and visualize this trade-off. I appreciate if the authors could expand on this topic, providing detailed explanations, methodologies, or visual aids to help readers understand the implications of this trade-off on the model's outputs? How does addressing this trade-off contribute to the robustness and fairness of the model?

---

### Official Review · Reviewer_w6Hu · 2023-11-06

**Soundness:** 3 good
**Presentation:** 2 fair
**Contribution:** 1 poor
**Rating:** 3
**Confidence:** 3

**Summary:**

The paper proposes a geometric framework for fairness, where distributions and (learned) conditional distributions are represented as infinite-dimensional vectors, and hypothesis classes as sets. It makes the observation that a model that maximizes accuracy based on observed labels may not always be optimal for fairness.

**Strengths:**

The paper is clear and I see no issue with the claims made.

**Weaknesses:**

In my view, the core observation of the paper is that a distribution can be represented as an infinite length transcript of samples from that distribution, and function evaluations can likewise be represented as the infinite length vector of the evaluation of the function over the transcript. This representation conserves conditional means (in the almost sure sense).

I just fail to see how this framework provides any insight, actionable or otherwise, on the fairness problem that is not equally or more simply understood in the original statistical framework. Figure 1, for example, could just as easily be drawn using the set of achievable (conditional) probability distributions.

Definitions 1 and 2 are additionally 'intuitive' but not formal.

**Questions:**

What are the practical or theoretical advantages of this framework over the standard probability framework.

---

### Author Response · Authors · 2023-11-14
**Thank you for your consideration**

I am writing to express my gratitude for the opportunity to submit our paper. I appreciate the time and effort invested by the conference committee in the review process.

While I am naturally disappointed to learn that my paper did not meet the criteria for acceptance, I value the feedback provided by the reviewers. Their insights are invaluable in helping us understand the shortcomings of my work and providing guidance for potential improvements.

I understand the rigorous standards that the ICLR upholds, and I appreciate the committee's commitment to maintaining the quality of the conference proceedings. We remain committed to contributing to the creation of a complete geometric framework for fairness and will use the feedback received to strengthen the scholarly merit of the research.
Once again, I thank the committee for the opportunity to participate in the review process. I look forward to future opportunities to engage with the ICLR.

Thank you for your time and consideration.

---

### Meta-Review · Area_Chair_W4qY · 2023-12-12

**Metareview:**

This paper takes a novel approach to presenting fairness analyses (train- and inference-time) in a more intuitive fashion.  Reviewers appreciated the motivation for the approach but found critical issues with its current implementation and presentation.  Reviewers and this AC see this as a strong motivation for an idea that can be curated into a future top-level publication.

**Justification For Why Not Higher Score:**

No rebuttal. Negative reviews.

**Justification For Why Not Lower Score:**

N/A

---

### Decision · Program_Chairs · 2024-01-16

Reject